# TaKF$^+$: A versatile and parameter-efficient tuning for EEG foundation model

## Abstract

Electroencephalogram (EEG) data, widely used in brain-computer interfaces (BCIs), pose challenges for reusing deep learning models trained on specific datasets due to variations in recording configurations and domain gaps. While foundation models pre-trained on large-scale EEG datasets have emerged as a promising solution, the challenge of effectively adapting them to downstream tasks has yet to be fully explored. To address this, we propose a novel tuning method, TaKF$^+$, which consists of the Task-Adaptive Key-Feature Extractor (TaKF) and adapter modules. TaKF$^+$ is designed to efficiently extract task-relevant features from EEG foundation models for downstream tasks while preserving the model's parameters and significantly reducing computational overhead. We evaluate TaKF$^+$ across a diverse range of tasks, including motor imagery, emotion recognition, and seizure detection, and demonstrate its superior performance and adaptability compared to existing methods over publicly available datasets. Our research paves the way for more efficient and versatile applications of EEG foundation models across various domains.

## 1 Introduction

An electroencephalogram (EEG) is a record of changes in the membrane potential, which is obtained from electrodes placed on the scalp surface. EEG is widely used in brain-computer interfaces (BCIs) due to its non-invasive nature and ability to capture meaningful information related to neural activity, making it a valuable tool for interpreting brain states (Phyo et al., 2022; Kim et al., 2024; Panda et al., 2010). Recently, deep learning has become a standard for BCI-based EEG analysis with remarkable performance. Specifically, utilizing neural networks with supervised learning can significantly enhance the performances of diverse EEG tasks by capturing the inherent patterns of EEG (Lawhern et al., 2018; Song et al., 2022). However, despite the existence of the standard international 10-20 system for EEG data recording, heterogeneity in configurations—such as the number of channels, electrode placements, and sampling rates—across different measurement institutions creates a domain gap (Dornhege et al., 2004; Brunner et al., 2008). Additionally, there is also a gap caused by the high variability between subjects and datasets (Ko et al., 2022). These gaps ultimately limit the generalizability of dataset-specific trained models. In practice, these challenges result in limitations, such as the need for large amounts of data and increased computational costs, as neural networks often need to be retrained for new tasks or different patients to maintain performance.

Transfer learning-based deep learning models for EEG have been introduced to mitigate these challenges (Zhang et al., 2017). To enhance the generalization of models across different datasets, transfer learning utilizes prior knowledge of models, resulting in reduced dependency on large amounts of task-specific data and lower computational cost associated with retraining neural networks for new tasks or subjects (Cai et al., 2023). Generally, through the pre-training process, transfer learning acquires general representations that can be applied across subjects (Ko et al., 2024). In particular, the foundation model, a pretrained neural network that has learned general features from large datasets for various downstream tasks, can be a powerful breakthrough (Zhou et al., 2023). With this in mind, attempts have been made to develop transformer-based EEG foundation models using masked prediction as a pretext task (Yang et al., 2024; Zhang et al., 2024; Jiang et al., 2024). These efforts overcome the heterogeneities in configurations and dissolve inter-dataset variability, pointing towards developing EEG foundation models that can be easily used for new tasks.

However, while research on developing effective pretext tasks for EEG foundation model pretraining is steadily increasing, insufficient consideration has been given to how these EEG foundation models can be applied to downstream tasks. Existing EEG foundation models perform downstream tasks by fully fine-tuning all their parameters, which inevitably entails high computational costs and can lead to a degradation of generalization ability (Kumar et al., 2022). Parameter-efficient fine-tuning (PEFT) methods, which adapt the foundation model to new tasks by modifying only a tiny portion of parameters, offer a potential solution (Zaken et al., 2021). Especially in clinical applications, EEG foundation models must be adapted to specific patients due to the crucial importance of stability and accuracy. In these situations, additive fine-tuning, which updates only additional modules without altering the parameters of the foundation model (Han et al., 2024), is well-suited in terms of

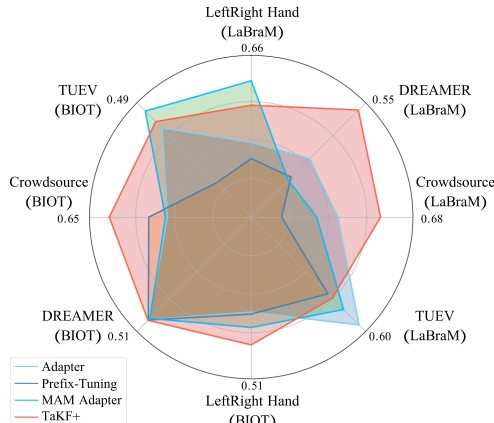

Figure 1: The overall performance comparison of TaKF$^+$ and other additive tuning methods across diverse EEG downstream tasks.

storage and financial costs. However, our analysis shows two limitations when applying additive fine-tuning methods to EEG foundation models. One limitation is, as shown in Figure 1, the inconsistency in performance that occurs depending on the EEG foundation model or the downstream task. In particular, these methods show a large dependency on which datasets are used for EEG foundation model pre-training. Another limitation is poor few-shot performance, a critical issue given the difficulty obtaining sufficient EEG data.

Therefore, in this paper, we propose a novel additive fine-tuning method, TaKF$^+$, which can be applied task-agnostically and is robust in few-shot scenarios. To be a versatile tool for diverse downstream tasks, TaKF$^+$ is divided into two parts: the Task-Adaptive Key-Feature Extractor (TaKF), which enhances the neural network's expressiveness, and the adapter modules, which modify the prior knowledge of the EEG foundation model in a task-specific manner. Notably, TaKF has the advantage of data efficiency by extracting task-relevant features in a lower-dimensional space than the original dimensions of the EEG foundation model. We evaluate the effectiveness of our proposed method on a wide range of downstream tasks, such as seizure detection, emotion recognition, and motor imagery. Consequently, we demonstrate that our proposed method can serve as a versatile and task-agnostic tool and excels in few-shot scenarios. The contributions of this work are summarized as follows:

- We propose a novel additive fine-tuning method, called TaKF$^+$, specifically designed to obtain task-relevant key features from two complementary views through the TaKF and adapter modules, which can provide a versatile solution for a broad spectrum of downstream tasks with EEG.

- We achieve significant improvements in data-scarce environments and mitigate dependency on labeled data by proposing the TaKF module, which operates in a low-dimensional space using learnable low-dimensional queries.

- We validate the effectiveness and efficiency of the proposed method through extensive experimentation, demonstrating its superior performance across various applications, including seizure detection, emotion recognition, and motor imagery over four publicly available datasets.

## 2 RELATED WORKS

### 2.1 FOUNDATION MODELS FOR EEG SIGNALS

Foundation models are pre-trained neural networks with large datasets to learn general representation, so that they can work well in a wide range of downstream tasks (Zhou et al., 2023). The remarkable efficacy of foundation models in enhancing EEG-based BCI systems has especially garnered

notable attention. Kostas et al. (2021) developed a universal pre-trained model, BENDR, trained using contrastive self-supervised learning, for wider EEG-based analysis. Cai et al. (2023) suggested a self-supervised learning framework that can learn implicit spatial and temporal correlations through pretext tasks reflecting the characteristics of brain signals. Kostas et al. (2021) and Cai et al. (2023) attempted to advance towards large models by training on massive-sized source datasets, but they did not address diverse configurations of EEG signals. Wang et al. (2023) proposed a reusable model named BrainBERT, pre-trained using a masking strategy, to provide embeddings for intracranial recordings. Brant, devised by Zhang et al. (2024), is a general and large-scale model that learns the long-term temporal dependency and spatial correlation of intracranial neural signals for adaption across a wide range of tasks. Brant and BrainBERT are designed to handle signals with different numbers of channels using a single model, but they were pre-trained on only a single dataset. Yang et al. (2024) demonstrated that the BIOT model, trained on multiple datasets with different formats, performs well across a wide range of downstream tasks. The foundation model proposed by Jiang et al. (2024), LaBram, trained via self-supervised learning using a tokenizer on tremendous EEG data, exhibited excellent performance in out-of-source datasets. On the other hand, by pointing out the weaknesses of the masked prediction pretext task, Foumani et al. (2024) introduces a self-prediction approach, EEG2Rep, which enables the generation of rich semantic representations. Despite these advancements, there have been few in-depth discussions on how to effectively apply these models to diverse downstream tasks.

## 2.2 PARAMETER-EFFICIENT FINE-TUNING

Fine-tuning pre-trained models for specific tasks is a common and effective way to enhance learning by leveraging knowledge from related domains (Yosinski et al., 2014; Ko et al., 2024). However, as the parameter scale of pre-trained models continues to grow (Kaplan et al., 2020), the inefficiencies and costs associated with fine-tuning have been pointed out as critical issues (He et al., 2021). Recently, PEFT, which updates only a tiny proportion of parameters while freezing the rest of the pre-trained model, has emerged as a solution to this issue (Houlsby et al., 2019; Zaken et al., 2021). Among the proposed PEFT methods, additive fine-tuning algorithms adopt additional trainable modules and only tune these added modules while keeping the pre-trained model unchanged (Han et al., 2024). The adapter-form approach, initially introduced by the Adapter (Houlsby et al., 2019), is the most well-known concept among additive fine-tuning algorithms. An adapter module, used in the adapter-form approach, plays a key role in encoding the representations in the intermediate layers into a task-specific form. Alternatively, the soft prompt approach improves performance through the continuous embedding space (Petrov et al., 2023). Li & Liang (2021) introduces learnable vectors that are prepended to the keys and values in the multi-head attention mechanism of transformer blocks, making models more expressive. Jia et al. (2022) demonstrates that the prefix-tuning method is adaptable to transformer-based vision models. Apart from these, He et al. (2021) explores the connections between additive fine-tuning approaches and proposes a hybrid form known as the MAM Adapter, which integrates Adapter and Prefix-Tuning. However, despite many advancements, these approaches have not yet been validated for EEG-related pre-trained models or their suitability for BCI tasks.

## 3 PRELIMINARY

### 3.1 ADAPTER-FORM APPROACH

The adapter-form approach (Houlsby et al., 2019) employs a trainable module (adapter) which is inserted in transformer layers. The adapter generally consists of a down-projection matrix $W_{\text{down}} \in \mathbb{R}^{d \times r}$, followed by a non-linear activation function $\sigma(\cdot)$, and an up-projection matrix $W_{\text{up}} \in \mathbb{R}^{r \times d}$, which first projects the input vector $h$ to a lower-dimensional space of size $r$, then projects it back to the original dimension $d$. The adapter process operates as:

$$h \leftarrow h + \sigma(h W_{\text{down}}) W_{\text{up}}. \tag{1}$$

These methods utilizing adapter modules have been shown in the natural language processing (NLP) field to achieve comparable performance to full fine-tuning with only a few tunable parameters. Depending on the purpose, the architecture and usage of adapters can be modified, as seen in Parallel Adapter (He et al., 2021), AdaMix (Wang et al., 2022), and AdapterSoup (Chronopoulou et al., 2023).

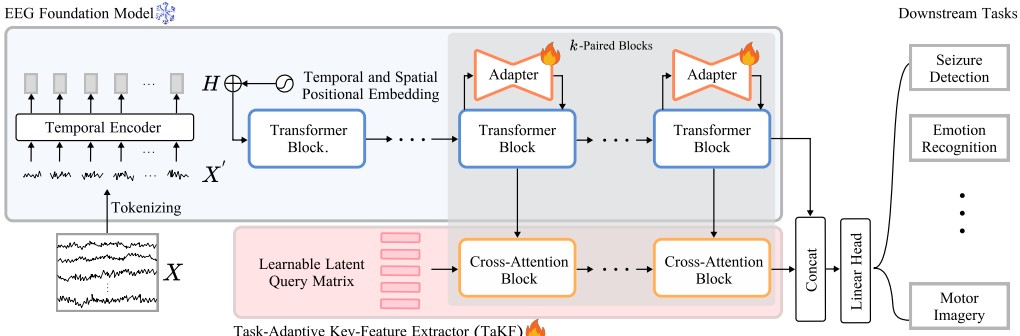

Figure 2: Overview of the proposed framework, TaKF$^+$. TaKF$^+$ includes the EEG foundation model, the Task-Adaptive Key-Feature Extractor (TaKF), and adapter modules. Only the parameters of the TaKF and adapter modules are updated during downstream tasks, while the EEG foundation model remains frozen during fine-tuning

### 3.2 PREFIX-TUNING

Motivated by the progress of prompt-based learning methods in the NLP field, prefix-tuning (Li & Liang, 2021) introduces $n$ prefix vectors, which have a dimension of $d$, and concatenates (Concat) them with keys and values in multi-head attention (MHA) layer. Specifically, the $n$ prefix vectors for the key, $\boldsymbol{P}_k \in \mathbb{R}^{n \times d}$, and for the value, $\boldsymbol{P}_v \in \mathbb{R}^{n \times d}$, are appended to the original key $\boldsymbol{K}$ and value $\boldsymbol{V}$, either at the first or at every attention (Attn) layer (Jia et al., 2022). Then, multi-head attention is applied to the concatenated keys and values, with the computation for each head being as follows:

$$\text{MHA}(\boldsymbol{C}, \boldsymbol{x}, \boldsymbol{P}_k, \boldsymbol{P}_v) = \text{Concat}(\text{head}_1, \dots, \text{head}_h)\boldsymbol{W}_o, \tag{2}$$

$$\text{head}_i = \text{Attn}(\boldsymbol{x}\boldsymbol{W}_q^{(i)}, \text{Concat}(\boldsymbol{P}_k^{(i)}, \boldsymbol{C}\boldsymbol{W}_k^{(i)}), \text{Concat}(\boldsymbol{P}_v^{(i)}, \boldsymbol{C}\boldsymbol{W}_v^{(i)})), \tag{3}$$

where $\boldsymbol{W}_o \in \mathbb{R}^{d \times d}$, $\boldsymbol{W}_q^{(i)}, \boldsymbol{W}_k^{(i)}, \boldsymbol{W}_v^{(i)} \in \mathbb{R}^{d \times d/N_d}$ denotes projection matrix of query, key, and value, $\boldsymbol{C} \in \mathbb{R}^{m \times d}$ represents a given sequence of $m$ vectors, $\boldsymbol{x} \in \mathbb{R}^d$ indicates a query vector, and head$_i$ refers to the $i$-th head's vectors, and $\boldsymbol{P}_k^{(i)}$ and $\boldsymbol{P}_v^{(i)} \in \mathbb{R}^{n \times d/N_d}$ correspond to the respective parts of $\boldsymbol{P}_k$ and $\boldsymbol{P}_v$. Notably, its superiority is particularly evident in low-data settings with a limited parameter budget.

## 4 METHOD

### 4.1 OVERALL FRAMEWORK

Given diverse downstream tasks $\mathcal{T} = \{\mathcal{T}_1, \mathcal{T}_2, \dots, \mathcal{T}_S\}$, which can correspond to specific datasets or subjects, we aim to address $\mathcal{T}$ by leveraging the EEG foundation model effectively. For a specific task $\mathcal{T}_s$, our framework is divided into a frozen EEG foundation model with pre-trained weights $\theta$ and additive tunable modules with learnable parameter $\phi_s$, where $\phi_s$ has a significantly smaller parameter size compared to $\theta$. We only update the learnable parameters $\phi_s$. Since the approach to adapting the EEG foundation model may vary depending on $\mathcal{T}_s$, we divise the additive tunable modules of two complementary components to reduce this variability: a TaKF, which makes the model more expressive for capturing new task-relevant patterns, and adapter modules, which transform the prior knowledge of the EEG foundation model into a task-specific form.

As shown in Figure 2, an input EEG signal $\boldsymbol{X} \in \mathcal{T}_s$ is passed through the EEG foundation model for decoding. The input EEG signal is embedded through two pathways: (1) transformer blocks combined with adapter modules and (2) the TaKF. The two features, embedded through separate pathways, are concatenated and used for prediction. This process updates the additive tunable parameters $\phi_s$. As a result, we can handle $\mathcal{T}$ by using the pre-trained weights $\theta$ and $\phi_1, \phi_2, \dots, \phi_S$, which are obtained from each downstream process.

## 4.2 EEG FOUNDATION MODEL

The EEG foundation model $f_\theta$, consisting of a patch embedding encoder and $L$ transformer encoder blocks, is a neural network that has learned EEG-related representations through pretext tasks, such as masked patch prediction (Zhou et al., 2023). The input EEG signal $\boldsymbol{X} \in \mathbb{R}^{c \times t}$, where $c$ represents the number of electrodes (channels) and $t$ represents the number of time steps in the signal, is tokenized by segmenting it into patches of size $1 \times w$, resulting in a reshaped input signal $\boldsymbol{X}' \in \mathbb{R}^{(c \times \lfloor t/w \rfloor) \times w}$. The segmented input is then encoded into $d$-dimensional tokens $\boldsymbol{H} \in \mathbb{R}^{(c \times \lfloor t/w \rfloor) \times d}$ using a temporal encoder. Temporal and spatial positional embedding vectors, as positional encodings (Vaswani et al., 2017), are added to these tokens. The tokens are then passed through $L$ transformer encoder blocks and concatenated with the output of the TaKF.

## 4.3 TASK-ADAPTIVE KEY-FEATURE EXTRACTOR

The TaKF aims to extract task-relevant information from the EEG foundation model and compress it into low-dimensional features. To perform this role, the TaKF utilizes cross-attention blocks to extract important features from the representation features and project them into the learnable latent query matrix. The TaKF has $J$ cross-attention blocks and a learnable latent query matrix $\boldsymbol{Q}_0 \in \mathbb{R}^{N \times r}$, which consists of $N$ learnable latent query vectors $\boldsymbol{q}_0 \in \mathbb{R}^{1 \times r}$. Each learnable latent query vector $\boldsymbol{q}_0$ is a randomly initialized $r$-dimensional learnable parameter, where $r$ is less than the dimension $d$ of the representation features of the EEG foundation model. The cross-attention blocks are paired with the last $J$ transformer blocks of the EEG foundation model, as high-level features are closely related to the targets, while low-level features capture common characteristics (Ren et al., 2023). The query matrix that passes through the $J$ cross-attention-based blocks is used as the input to the linear head.

**Cross-Attention Block.** The ability of the cross-attention to extract and compress large input information into a low-dimensional array has been validated in numerous works, including Perceiver (Jaegle et al., 2021) and Q-former (Li et al., 2023). Furthermore, cross-attention offers parameter-efficient advantages in mapping high-dimensional vectors to low-dimensional vectors. Therefore, we adopt the cross-attention for extracting task-relevant features from the large and high-dimensional representation features of the EEG foundation model.

As shown in Figure 3, the cross-attention block consists of a cross-attention and residual connection. The first cross-attention block uses the initial latent query matrix $\boldsymbol{Q}_0$, which is learnable and stored within the neural network, as queries for cross-attention. In the $j$-th cross-attention block, for $j > 0$, the latent query matrix $\boldsymbol{Q}_{(j-1)}$, which have passed through $(j-1)$ preceding cross-attention blocks, act as queries for cross-attention. For the keys and values in the cross-attention, the representation feature

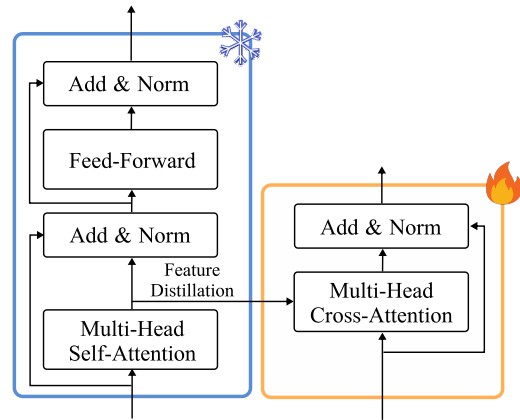

Figure 3: The illustration of a paired block in TaKF$^+$. The cross-attention block uses the output of self-attention within the paired transformer block as the key and value for cross-attention. In the figure, the adapter module attached to the transformer block was omitted.

$\boldsymbol{H}_l \in \mathbb{R}^{(c \times \lfloor t/w \rfloor) \times d}$, obtained from the $l$-th transformer block of the EEG foundation model, is used. In this case, the $l$-th transformer block is paired with the $j$-th cross-attention block. Before using the latent query matrix and the representation feature as queries, keys, and values, Layer Normalization (Ba, 2016) is applied to each to ensure training stability (Dehghani et al., 2023). After the cross-attention process, a residual connection (He et al., 2016) is applied to the output. The process

of $j$-th cross-attention (Cross-Attn) block can be described as follows:

$$\text{Cross-Attn}(\boldsymbol{Q}_{(j-1)}, \boldsymbol{H}_l) = \text{Softmax}\left(\frac{\text{LN}(\boldsymbol{Q}_{(j-1)})\boldsymbol{W}_q \cdot (\text{LN}(\boldsymbol{H}_l)\boldsymbol{W}_k)^T}{\sqrt{r}}\right)\text{LN}(\boldsymbol{H}_l)\boldsymbol{W}_v, \quad (4)$$

$$\boldsymbol{Q}_j = \boldsymbol{Q}_{(j-1)} + \text{Cross-Attn}(\boldsymbol{Q}_{(j-1)}, \boldsymbol{H}_l), \quad (5)$$

where LN($\cdot$) denotes Layer Normalization, and $\boldsymbol{W}_q \in \mathbb{R}^{r \times r}$, $\boldsymbol{W}_k \in \mathbb{R}^{d \times r}$, and $\boldsymbol{W}_v \in \mathbb{R}^{d \times r}$ are the query, key, and value projection matrices, respectively.

**Feature Distillation from Before the Feed-forward Layer.** When passing information from the EEG foundation model to the TaKF, we used the representation features taken from before the feed-forward layer of the transformer block in the EEG foundation model. We aimed to find a way for the EEG foundation model to pass information to the TaKF module to effectively capture task-relevant features for tasks that are not closely related to the prior knowledge of the EEG foundation model. Our analysis confirmed that the performance of TaKF varies significantly depending on whether the representation features are taken before or after the feed-forward layer, as detailed in Appendix C. Therefore, to maximize the intended functionality of TaKF, we set the feature distillation position.

### 4.4 ADAPTER MODULE

For tasks closely related to the pre-training dataset, the EEG foundation model already contains sufficient relevant information, while capturing task-relevant features through the TaKF module can be highly effective for tasks not directly associated with the pre-training dataset. In such cases, the adapter, which transforms the representations of intermediate layers in transformer blocks into task-specific features (Houlsby et al., 2019), is a powerful tool. Therefore, we integrated adapter modules into our framework. Inspired by Mix-adapter (He et al., 2021), we use the adapter module exclusively in parallel with the feed-forward layer of the transformer blocks.

## 5 EXPERIMENT

### 5.1 DATASETS

We employ four EEG datasets to validate our proposal. 1) **Left/Right Hand Motor Imagery (Left-Right Hand)** is a binary dataset with two labels: left-hand and right-hand (Zakrzewski et al., 2022). 2) **DREAMER** (Katsigiannis & Ramzan, 2017) contains signals related to a subject's affective state, including values for valence, arousal, and dominance. 3) **Crowdsourced** is a binary dataset (eyes open vs. eyes closed) from the EMOTIV platform (Williams et al., 2023). 4) **TUEV** Obeid & Picone (2016) contains signals classified into six abnormal-related events. Details of the datasets is described in Appendix A.1. The code can be accessed on Github[1].

### 5.2 PREPROCESSING AND EVALUATION METRICS

We follow essential preprocessing steps from LaBraM (Jiang et al., 2024). Further details are provided in the Appendix A.2. For the data division, we followed the approach described below: 1) **DREAMER**, **LeftRight Hand**, and **Crowdsourced**: we evaluate them using 5-fold cross-validation based on subjects. 2) **TUEV**: The division for training and test sets is provided by the dataset. For the validation group, we split the training set into an 80%:20% ratio based on subjects. We use balanced accuracy (BACC) and the area under the ROC curve (AUROC) as evaluation metrics for binary classification datasets. For multi-class tasks, we adapt balanced accuracy (BACC) and Cohen's kappa (Cohen's $\kappa$).

### 5.3 BASELINE

We evaluate our method using three categories: 1) supervised modeling methods (SMM), 2) self-supervised modeling methods (Self-SMM), and 3) additive fine-tuning methods. The supervised modeling methods, which are simply fine-tuned on each downstream dataset, include SPaRCNet

---

[1]https://anonymous.4open.science/r/TaKFplus

Table 1: Performance comparison to the competitors on LeftRight Hand, DREAMER, Crowd-sourced, and TUEV datasets, using LaBraM as the base EEG foundation model. FT and LP denote fine-tuning and linear probing, respectively, while PT represents Prefix-Tuning. Bold values represent the best results, while underlined values indicate the second-best.

| | **LeftRight Hand** | | **DREAMER** | |
|---|---|---|---|---|
| | BACC | AUROC | BACC | AUROC |
| SMM SOTA | $69.06 \pm 5.16$ | $77.75 \pm 6.02$ | $56.25 \pm 4.10$ | $59.04 \pm 7.17$ |
| Self-SMM SOTA | $60.26 \pm 5.79$ | $66.34 \pm 8.18$ | $55.63 \pm 2.85$ | $58.02 \pm 2.74$ |
| LaBraM-FT (Jiang et al., 2024) | $70.01 \pm 4.36$ | $77.71 \pm 5.73$ | $55.67 \pm 3.64$ | $59.60 \pm 4.79$ |
| LaBraM-LP | $52.06 \pm 2.10$ | $53.83 \pm 3.07$ | $50.51 \pm 1.13$ | $53.20 \pm 3.23$ |
| LaBraM-Adapter (Houlsby et al., 2019) | $62.91 \pm 4.83$ | $70.72 \pm 7.58$ | $\underline{53.61 \pm 3.48}$ | $\underline{57.47 \pm 6.68}$ |
| LaBraM-PT (Li & Liang, 2021) | $62.28 \pm 3.11$ | $68.88 \pm 4.75$ | $53.10 \pm 3.01$ | $57.10 \pm 3.90$ |
| LaBraM-MAM Adapter (He et al., 2021) | $\mathbf{65.31 \pm 4.74}$ | $\mathbf{73.02 \pm 6.99}$ | $53.02 \pm 3.32$ | $54.85 \pm 3.94$ |
| (Ours) LaBraM-TaKF$^+$ | $\underline{64.36 \pm 5.66}$ | $\underline{71.66 \pm 7.72}$ | $\mathbf{54.95 \pm 3.84}$ | $\mathbf{59.31 \pm 4.86}$ |
| | **Crowdsourced** | | **TUEV** | |
| | BACC | AUROC | BACC | Cohen's $\kappa$ |
| SMM SOTA | $60.71 \pm 11.99$ | $74.99 \pm 9.32$ | $43.84 \pm 3.49$ | $39.12 \pm 2.37$ |
| Self-SMM SOTA | $65.77 \pm 12.84$ | $69.27 \pm 3.08$ | $53.37 \pm 1.10$ | $52.61 \pm 2.44$ |
| LaBraM-FT | $62.04 \pm 10.51$ | $65.30 \pm 11.15$ | $64.09 \pm 0.65$ | $66.37 \pm 0.93$ |
| LaBraM-LP | $54.95 \pm 3.23$ | $68.11 \pm 7.67$ | $34.61 \pm 2.25$ | $39.68 \pm 3.29$ |
| LaBraM-Adapter | $\underline{65.37 \pm 11.17}$ | $\underline{74.75 \pm 5.89}$ | $\mathbf{59.86 \pm 0.98}$ | $\mathbf{56.88 \pm 1.52}$ |
| LaBraM-Prefix | $63.18 \pm 10.45$ | $72.06 \pm 14.11$ | $55.56 \pm 1.94$ | $52.44 \pm 3.48$ |
| LaBraM-MAM Adapter | $64.55 \pm 9.87$ | $71.93 \pm 14.79$ | $\underline{57.73 \pm 0.59}$ | $51.72 \pm 2.05$ |
| (Ours) LaBraM-TaKF$^+$ | $\mathbf{67.04 \pm 14.20}$ | $\mathbf{75.46 \pm 12.74}$ | $56.17 \pm 1.45$ | $\underline{54.27 \pm 1.14}$ |

(Jing et al., 2023), ContraWR (Yang et al., 2023), CNN-Transformer (Peh et al., 2022), FFCL (Li et al., 2022), and ST-Transformer (Song et al., 2021). Self-supervised modeling methods, which first undergo pre-training to learn semantic representations from EEG data followed by fine-tuning, include BIOT (Yang et al., 2024), EEG2Rep (Foumani et al., 2024), and LaBraM (Jiang et al., 2024). Although EEG2Rep is capable of cross-domain transfer learning, we evaluate them solely on in-domain tasks to ensure consistency in comparison and due to limitations in channel configurations. BIOT and LaBraM are evaluated using the released pre-trained weights. The final category, additive fine-tuning methods, includes Adapter (Houlsby et al., 2019), Prefix-Tuning (Li & Liang, 2021), and MAM Adapter (He et al., 2021). We evaluate these methods using an equal tunable parameter ratio of 3% and the same EEG foundation model with the publicly available parameters. To demonstrate the effectiveness of our method across EEG foundation models, we adopt two EEG foundation models, LaBraM and BIOT. Details about baseline models are described in Appendix A.4.

## 6 EXPERIMENTAL RESULTS

### 6.1 MAIN RESULTS

We present the summary results in Tables 1 and 2. In the analysis, while additive fine-tuning methods entirely depend on the potential of the EEG foundation model, direct comparisons between the proposed and baseline methods, such as supervised and self-supervised modeling methods, are not necessarily fair. Instead, it is important to note that both the proposed and existing additive fine-tuning methods guarantee performance comparable to the EEG foundation model or maintain the EEG foundation model's outperformance compared to other baseline methods. To this end, we primarily compare our proposal with the additive fine-tuning methods. We briefly present the state-of-the-art (SOTA) for both SMMs and Self-SMMs separately in the tables, with detailed results in Appendix B. Bold values represent the best results, while underlined values indicate the second-best, both shown only within the context of additive fine-tuning results without fine-tuning.

Table 2: Performance comparison to the competitors on LeftRight Hand, DREAMER, Crowd-sourced, and TUEV datasets, using BIOT as the base EEG foundation model. FT and LP denote fine-tuning and linear probing, respectively, while PT represents Prefix-Tuning. Bold values represent the best results, while underlined values indicate the second-best.

| | LeftRight Hand | | DREAMER | |
| --- | --- | --- | --- | --- |
| | BACC | AUROC | BACC | AUROC |
| SMM SOTA | $69.06 \pm 5.16$ | $77.75 \pm 6.02$ | $56.25 \pm 4.10$ | $59.04 \pm 7.17$ |
| Self-SMM SOTA | $60.74 \pm 3.51$ | $41.44 \pm 5.32$ | $55.67 \pm 3.64$ | $59.60 \pm 4.79$ |
| BIOT-FT (Yang et al., 2024) | $49.32 \pm 0.70$ | $49.55 \pm 0.91$ | $49.04 \pm 1.94$ | $48.88 \pm 3.13$ |
| BIOT-LP | $50.11 \pm 0.61$ | $50.53 \pm 1.09$ | $49.78 \pm 0.95$ | $49.78 \pm 2.66$ |
| BIOT-Adapter (Houlsby et al., 2019) | $49.85 \pm 0.90$ | $49.81 \pm 1.46$ | $50.35 \pm 1.73$ | $49.32 \pm 2.35$ |
| BIOT-PT (Li & Liang, 2021) | $49.89 \pm 0.43$ | $\underline{51.53 \pm 0.69}$ | $50.40 \pm 1.31$ | $49.57 \pm 1.69$ |
| BIOT-MAM Adapter (He et al., 2021) | $\underline{50.15 \pm 0.42}$ | $\mathbf{51.81 \pm 1.51}$ | $50.23 \pm 1.38$ | $\mathbf{51.49 \pm 3.51}$ |
| (Ours) BIOT-TaKF$^+$ | $\mathbf{50.49 \pm 0.75}$ | $50.61 \pm 0.87$ | $\mathbf{50.43 \pm 1.55}$ | $\underline{49.92 \pm 1.20}$ |
| | Crowdsourced | | TUEV | |
| | BACC | AUROC | BACC | Cohen's $\kappa$ |
| SMM SOTA | $60.71 \pm 11.99$ | $74.99 \pm 9.32$ | $43.84 \pm 3.49$ | $39.12 \pm 2.37$ |
| Self-SMM SOTA | $62.04 \pm 10.51$ | $65.30 \pm 11.15$ | $64.09 \pm 0.65$ | $66.37 \pm 0.93$ |
| BIOT-FT | $57.71 \pm 8.11$ | $69.42 \pm 9.39$ | $52.81 \pm 2.25$ | $52.73 \pm 2.49$ |
| BIOT-LP | $\underline{61.87 \pm 8.16}$ | $68.62 \pm 9.17$ | $37.47 \pm 1.25$ | $46.66 \pm 2.48$ |
| BIOT-Adapter | $58.16 \pm 8.24$ | $65.44 \pm 5.70$ | $45.54 \pm 2.77$ | $\underline{51.12 \pm 4.33}$ |
| BIOT-Prefix | $59.98 \pm 7.56$ | $\mathbf{72.55 \pm 8.30}$ | $36.01 \pm 1.45$ | $35.09 \pm 2.67$ |
| BIOT-MAM Adapter | $58.36 \pm 7.76$ | $70.08 \pm 6.90$ | $\mathbf{48.49 \pm 1.83}$ | $47.58 \pm 3.46$ |
| (Ours) BIOT-TaKF$^+$ | $\mathbf{63.83 \pm 7.06}$ | $\underline{70.44 \pm 4.70}$ | $\underline{46.66 \pm 1.22}$ | $\mathbf{51.56 \pm 2.03}$ |

**LaBraM Case.** Table 1 presents the results of using LaBraM as the EEG foundation model for applying additive fine-tuning methods. The results show that TaKF$^+$ outperformed or delivered performance comparable to other methods across most tasks. Especially in the Crowdsourced dataset, we observed that TaKF$^+$ achieved better performance than fine-tuning. It is notable that while other additive fine-tuning methods exhibit high variability depending on the dataset, our method maintained low variability across task types. Although the Adapter performed more stably than other baselines, it did not achieve the versatility of TaKF$^+$. Consequently, TaKF$^+$ is a versatile and effective tuning method when applied to LaBraM.

**BIOT Case.** Although BIOT is pre-trained on multiple datasets with a pretext task, it is noteworthy that BIOT is smaller, with only 3.3 million parameters, and is pre-trained on fewer datasets, specifically related to sleep and seizure, compared to LaBraM. The results of applying additive fine-tuning methods to BIOT are reported in Table 2. The results show that TaKF$^+$ achieves improved performance than other baselines. While Prefix-Tuning and MAM Adapter show better performance than Adapter due to their ability to enhance the expressiveness of the neural network, TaKF$^+$ surpasses them with a single setting across most datasets. Moreover, while fine-tuning underperforms except on non-seizure datasets due to the restricted prior knowledge of BIOT, TaKF$^+$ improves performance beyond fine-tuning. In particular, BIOT-TaKF$^+$ surpasses the Self-SMM SOTA, which is the case for LaBraM-FT, in the Crowdsourced dataset. As a result, This result demonstrates that TaKF$^+$ also functions as an effective additive fine-tuning method on BIOT.

**Compare Two Cases** In both cases, TaKF$^+$ performs well in terms of performance and variability compared to other additive fine-tuning methods. Specifically, in the case of LaBraM, the Adapter shows better stability than other baselines, likely due to LaBraM's potential, as it contains abundant information from diverse datasets. In contrast, the Adapter's ability is completely underutilized for BIOT, which has a narrower prior knowledge base. On the other hand, TaKF$^+$ consistently shows stable performance regardless of the EEG foundation model. Furthermore, the architecture of TaKF$^+$ is more suitable for EEG foundation models than the MAM Adapter. This difference

likely arises from using TaKF, which operates in a low-dimensional space, in contrast to the MAM Adapter, which uses Prefix-Tuning to make the neural network more expressive. In conclusion, given the absence of an additive tuning method that can be broadly applied to a wide range of downstream tasks in the BCI field, we believe that TaKF$^+$ could be a versatile and effective solution.

## 6.2 Ablation on Few-Shot Learning.

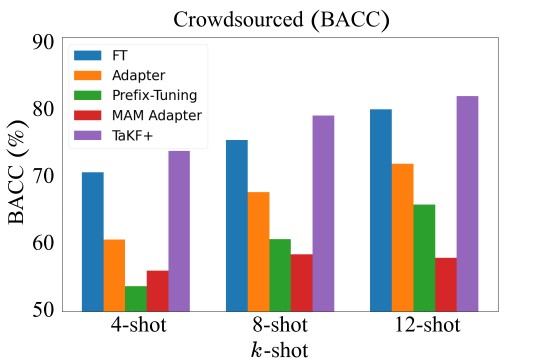 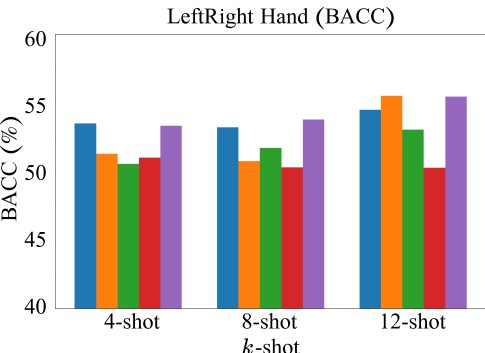

Figure 4: A comparison of the performance between our proposed method and existing additive fine-tuning methods under different few-shot settings (4-shot, 8-shot, and 12-shot). FT in the figure denotes fine-tuning.

While collecting a large amount of labeled EEG data can be expensive and time-consuming, data efficiency is a crucial issue (Ahuja & Sethia, 2024). To verify the data efficiency of each tuning method, we evaluate the performance of our proposed method and existing additive fine-tuning methods under an extremely limited training budget. We compare the methods across three few-shot settings (4-shot, 8-shot, and 12-shot) in a subject-dependent scenario using the Crowdsourced and LeftRight Hand datasets. The detailed numerical results can be found in the Appendix D. The results, illustrated in Figure 4, show that TaKF$^+$ is comparable to or better than other methods, including fine-tuning, in few-shot scenarios. Considering the comparison with the MAM Adapter, extracting task-relevant features in a low-dimensional space through introducing the TaKF module likely plays a key role. As a result, this verifies that TaKF$^+$ is strong in low-data regimes and has high data efficiency.

## 6.3 Ablation on Additive Tunable Modules.

Table 3: Ablation study of additive tunable modules on LaBraM. Bold values represent the best results, while underlined values indicate the second-best.

| | TUEV | | DREAMER | | Crowdsourced | |
|---|---|---|---|---|---|---|
| | BACC | Cohen's $\kappa$ | BACC | AUROC | BACC | AUROC |
| TaKF$^+$ | $\underline{56.17 \pm 1.45}$ | $\underline{54.27 \pm 1.14}$ | $\underline{54.95 \pm 3.84}$ | $\underline{59.31 \pm 4.86}$ | $\mathbf{67.04 \pm 14.20}$ | $\mathbf{75.46 \pm 12.74}$ |
| PA(FF) | $41.13 \pm 1.65$ | $46.00 \pm 1.85$ | $52.21 \pm 3.44$ | $55.98 \pm 4.55$ | $60.51 \pm 10.96$ | $71.82 \pm 14.53$ |
| PA | $\mathbf{59.40 \pm 2.23}$ | $\mathbf{56.35 \pm 1.71}$ | $52.21 \pm 3.44$ | $55.98 \pm 4.55$ | $58.85 \pm 7.95$ | $69.10 \pm 8.78$ |
| TaKF | $52.98 \pm 2.03$ | $49.81 \pm 0.97$ | $53.38 \pm 2.53$ | $59.36 \pm 2.49$ | $\underline{65.97 \pm 10.05}$ | $\underline{74.96 \pm 12.75}$ |
| TaKF(+FF) | $53.21 \pm 2.44$ | $51.90 \pm 1.72$ | $\mathbf{56.85 \pm 4.29}$ | $\mathbf{60.01 \pm 5.52}$ | $62.30 \pm 8.22$ | $68.76 \pm 6.87$ |

We conduct an ablation study to verify the contribution of each additive tunable module. We use an equal tunable parameter ratio of 3%. To evaluate the contribution of the TaKF module, we introduce a new architecture, TaKF(+FF), which adds a feed-forward layer to the cross-attention block to preserve the low-dimensional nature of TaKF with a fixed tunable parameter ratio. PA(FF) is equivalent to the tuning method that excludes a TaKF module from TaKF$^+$. Since having adapter modules attached only to the feed-forward network may negatively impact performance evaluation

for comparison, Parallel Adapter (PA) (He et al., 2021), which adds additional adapter modules to the attention layer within a limited parameter budget, is included as a comparison target. Table 3 presents the comparison results on the TUEV, DREAMER, and Crowdsourced datasets. The results demonstrate that each additive tunable module performs well as intended. Specifically, while the PA outperforms in TUEV, the TaKF(+FF) demonstrates outstanding ability in DREAMER. As a result, TaKF$^+$ successfully fuses the advantages of both the adapter-form approach and the TaKF, achieving synergy.

## 7 DISCUSSION

**Limitations.** First of all, although we verify the superiority of TaKF$^+$ on LaBraM and BIOT, there remains a need to confirm its adaptability to other EEG foundation models. Secondly, we have provided an explanation and experimental data for the scenarios where the additive fine-tuning methods each perform well, but there is a lack of theoretical analysis to support this explanation. Lastly, there is room for improving synergy considering the Table 3.

**Future works.** Considering the above limitations, we pave the way for future research directions. 1) Investigate enhancing the synergy or developing architectures tailored to the characteristics of EEG. 2) Develop the pre-training process by adapting additive fine-tuning methods to EEG foundation models that learn from different pretext tasks. Analyzing how prior knowledge is formed based on the pretext task through additive fine-tuning methods may provide key insights into the weaknesses of previous pre-training processes. 3) Discover the anatomical and physiological mechanisms of the brain by utilizing dataset- or subject-relevant key features extracted from TaKF$^+$.

## 8 CONCLUSION

We highlight the necessity of exploring methods to apply EEG foundation models to downstream tasks and identify areas for improvement by adapting existing methods from other domains. Based on this analysis, we propose a versatile tuning method, TaKF$^+$, which adapts the EEG foundation model to downstream tasks without altering the pre-trained weights. TaKF$^+$ is a tuning method that can be broadly applied across a wide range of downstream tasks, with a design that includes the TaKF module, which focuses on adaptive feature extraction for tasks, and adapter modules, which primarily leverage the prior knowledge of the EEG foundation model. Through experiments, we demonstrate that TaKF$^+$ achieves performance comparable to state-of-the-art methods for each dataset using a single architecture across most datasets. The proposed TaKF$^+$ presents a breakthrough in efficiently adapting EEG foundation models in a task-agnostic manner. Additionally, the ablation study verifies that TaKF$^+$ performs well in low-data scenarios and has the potential to significantly alleviate the challenges of sample and label efficiency in real-world medical applications, where available data is often limited.

ACKNOWLEDGMENTS

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
