# OpenReview forum: "TaKF$^{+}$: A versatile and parameter-efficient tuning for EEG foundation model"
_ICLR.cc/2025/Conference — ICLR 2025 Conference Withdrawn Submission_

### Official Review · Reviewer_KpEM · 2024-10-21

**Soundness:** 3
**Presentation:** 3
**Contribution:** 2
**Rating:** 6
**Confidence:** 4

**Summary:**

This paper focuses on reducing computational demands during the fine-tuning phase of large EEG models by training only newly added parameters. It introduces the TAKF method to enhance the model's expressiveness by extracting task-specific features, and incorporates an Adapter module to transfer foundational knowledge of the base model to specific tasks.

**Strengths:**

1.Innovation: Pioneering attention to the issue of high parameter counts during fine-tuning in large EEG models.
2.Significance: Fine-tuning usually requires adjusting all parameters, which can be computationally and temporally expensive. If the hypothesis holds, the corresponding optimizations could facilitate the widespread application of large models.
3.Clarity of writing: The descriptions of the proposed TAKF method and Adapter model are highlighted effectively.
4.Rich experimentation: A broad range of baseline comparisons including supervised and self-supervised learning SOTA methods were selected, and different approaches to fine-tuning with additional parameters were compared.
5.Reproducibility: The paper provides extensive code, and the reported results seem reproducible based on the documentation.

**Weaknesses:**

1.Innovation: In terms of methodology, only the TAKF module is newly introduced, while the Adapter model merely combines existing methods.
2.Notably unimpressive experimental results: As shown in Table 1, the performance on most datasets using LaBraM as the base model is significantly lower than LaBraM's fine-tuning results; intuitively, the lower computational cost may lead to a substantial decrease in effectiveness; on the TUEV dataset, it performs worse than the Adapter-only approach, which requires further analysis and explanation; according to Tables 1 and 2, it underperforms the MAM Adapter method in 3/8 of the metrics, showing no significant advantage.
3.Significant errors in tables: In the Appendix, Tables 7 and 8 present the same series of methods across four different datasets, yet the data for methods from LaBraM-LP to (Ours) LaBraM-TaKF+ are identical in both tables; there are also significant errors in table titles, e.g., Table 7 includes data for LeftRight Hand, which does not belong in the emotion recognition category. The authors are advised to carefully proofread the content.

**Questions:**

1. How much smaller is the amount of parameters added during the fine-tuning phase compared to the original model? Is it worth the potential reduction in effectiveness?
2. Both proposed modules increase trainable parameters to aid the fine-tuning process. Could they be demonstrated through interpretable methods, such as visualization, to substantiate the different effects described in the text?

---

### Official Review · Reviewer_urav · 2024-10-29

**Soundness:** 2
**Presentation:** 3
**Contribution:** 2
**Rating:** 5
**Confidence:** 4

**Summary:**

The paper presents TaKF+, a new approach for parameter-efficient fine-tuning of EEG foundation models. The Task-Adaptive Key-Feature Extractor (TaKF) combined with adapter modules enables efficient extraction of task-relevant features with minimal computational overhead, while maintaining or exceeding the performance of fully fine-tuned models in few-shot learning scenarios.

**Strengths:**

1. The proposed method maintains competitive performance while reducing computational cost by tuning only a small fraction of parameters, making it resource-efficient for real-world applications in EEG-based tasks.
2. The few-shot learning experiments demonstrate that TaKF+ approaches or even surpasses the performance of fully fine-tuned models in some datasets, which is a highly promising result.

**Weaknesses:**

1. While the paper states that 3% of parameters are tunable in additive fine-tuning methods, the exact tunable parameter ratio for TaKF+ is not provided. This lack of explicit comparison may lead to an unfair assessment of baseline methods. Clearly stating the tunable parameters for TaKF+ would provide a more transparent comparison.
2. The core idea of TaKF+—combining the well-established Adapter technique with a Q-former-like cross-attention mechanism—might be seen as a simple extension of known methods, limiting the novelty of the contribution.
3. The results indicate that TaKF+ does not consistently outperform all additive fine-tuning baselines across datasets. This inconsistency raises concerns about its general robustness and effectiveness.
4. Some widely used baselines, such as LoRA, Adaptformer, and UniPELT, are absent from the experimental comparison, limiting the comprehensiveness of the evaluation.
5. In Table 3, the performance of the proposed method's variants fluctuates significantly across different datasets, which casts doubt on the consistent effectiveness of individual components.

**Questions:**

1.  In Section 6.1, the paper mentions that "Although the Adapter performed more stably than other baselines, it did not achieve the versatility of TaKF+." What is meant by the versatility of TaKF+ in this context, and how is it quantitatively or qualitatively better than the Adapter in terms of versatility? More clarification is needed to justify this claim.

---

### Official Review · Reviewer_QXxs · 2024-11-02

**Soundness:** 2
**Presentation:** 2
**Contribution:** 2
**Rating:** 3
**Confidence:** 3

**Summary:**

The "TaKF+" paper presents a parameter-efficient tuning method aimed at enhancing EEG foundation models for diverse downstream tasks, such as seizure detection, emotion recognition, and motor imagery. The method, TaKF+, introduces a Task-Adaptive Key-Feature Extractor (TaKF) and adapter modules to adapt EEG foundation models in a task-agnostic manner, maintaining generalization and minimizing computational cost. Through evaluations on multiple datasets, the authors demonstrate TaKF+’s superior performance in few-shot scenarios and its adaptability across various EEG-based applications.

**Strengths:**

1.	TaKF+’s integration of the Task-Adaptive Key-Feature Extractor (TaKF) and adapter modules is novel and effective in tuning EEG foundation models with minimal parameter updates.
2.	The method is designed to work efficiently in low-data settings, demonstrating strong performance in few-shot learning scenarios.
3.	TaKF+ supports a broad range of downstream tasks, making it highly adaptable and suitable for diverse EEG-based applications.

**Weaknesses:**

1.	While TaKF+ introduces Task-Adaptive Key-Feature Extractor and adapter modules, the motivation behind this specific design choice seems not clear.  The paper lacks a more detailed comparison of how TaKF+ improves upon existing methods, both in terms of unique technical contributions and in addressing specific limitations of previous EEG foundation models
2.	The novelty of TaKF+ could be strengthened by discussing how it differs fundamentally from other parameter-efficient fine-tuning approaches beyond its application to EEG.
3.	Although the empirical results are promising, the paper needs a deeper theoretical rationale supporting the choice of parameter-efficient tuning for EEG foundation models. Specifically, a clearer explanation of why the TaKF+ structure is particularly suited for EEG data, as opposed to alternative architectures, would strengthen the paper’s foundation.
4.	Although TaKF+ shows improvement over some baselines, the paper should include more comparisons with recent advancements in EEG model tuning or transfer learning.
I will reconsider my assessment after checking authors' response

**Questions:**

1.	How does TaKF+ handle cases where downstream tasks have significantly different label distributions from the pre-trained EEG foundation model?
2.	Could the authors clarify how TaKF+ performs on larger, more heterogeneous EEG datasets that may have different sampling rates or noise levels?

---

### Official Review · Reviewer_QjAL · 2024-11-02

**Soundness:** 3
**Presentation:** 2
**Contribution:** 2
**Rating:** 3
**Confidence:** 4

**Summary:**

This paper introduces TaKF+, a parameter-efficient tuning method for adapting EEG foundation models to a variety of downstream tasks. TaKF+ combines a Task-Adaptive Key-Feature Extractor (TaKF) with adapter modules to extract and refine task-specific features while keeping the foundation model's parameters largely unchanged, thus minimizing computational costs. Through experiments on diverse EEG tasks like motor imagery and seizure detection, TaKF+ shows superior adaptability and stability compared to existing tuning methods, particularly in data-scarce settings. The study highlights TaKF+ as a versatile and efficient tool for EEG-based applications, addressing critical challenges in EEG model adaptation.

**Strengths:**

- Comprehensive related work situates TaKF+ well in EEG model adaptation literature.
- Tackling parameter-efficient tuning for EEG is timely and could make a significant impact if successful.
- TaKF+ is tested on 4 datasets and 2 recent pre-trained models.
- There is no overlap between the evaluation datasets and the training datasets used in the pre-trained models.

**Weaknesses:**

- I understand you want to show that TaKF+ is more robust than the baseline but the tables are hard to read: sometimes TaKF+ is better sometimes not. To show this, you could use normalized plots as in [1, 2].
- Heavy use of acronyms impacts readability: SMM, FT, LP, PT.
- An analysis of the performance with respect to the number of training samples would be interesting.
- A comparison of computational time with other methods would also be interesting.

[1] Mellot, A., Collas, A., Chevallier, S., Gramfort, A., & Engemann, D. A. (2024). Geodesic Optimization for Predictive Shift Adaptation on EEG data. arXiv preprint arXiv:2407.03878.

[2] Kim, M. J., Grinsztajn, L., & Varoquaux, G. (2024). CARTE: pretraining and transfer for tabular learning. ICML 2024.

**Questions:**

- What’s N_d in paragraph 3.2?
- What is a “self-supervised modeling method”?
- What is “SMM SOTA”? is it a neural network trained from scratch?

Typo:
- Eq (3): wrong matrix-vector shapes for “xW_q”

---

### Official Review · Reviewer_VxyC · 2024-11-03

**Soundness:** 2
**Presentation:** 3
**Contribution:** 2
**Rating:** 3
**Confidence:** 5

**Summary:**

In this article, the authors investigate fine-tuning techniques for pre-trained models in the context of EEG-based BCI. They present a method which combines adding adapter layers to the transformer  (adapter form approach) and learning additional vectors which are concatenated to the key and value vectors in the transformer (prefix-finetuning). Both approaches are used in order to reduce the number of parameters to finetune.

**Strengths:**

As pointed out by the authors, fine-tuning strategies are relatively underexplored with EEG foundation models. They start to fill this gap by proposing a novel fine-tuning algorithm.
The paper is well-structured and easy to follow, with good-quality figures. The diagrams are clear, and the use of pictograms makes their understanding intuitive.

**Weaknesses:**

- On the DREAMER and motor imagery (MI) datasets, the method proposed by the authors consistently produces relatively low results, underperforming compared to the supervised baseline. Lines 370-373, the authors suggest that this comparison is not fair and only compare the adaptive methods within themselves. However, I respectfully disagree and maintain that it is indeed appropriate and relevant. Indeed, all models had access to the same quantity of data from the target distribution. The fact that the pre-trained models perform poorly means that they are not able to correctly use the available target data. This issue is called “negative transfer” and it needs to be tackled, not ignored.
- The models based on BIOT systematically perform around chance-level (50±2%) on the MI and DREAMER datasets. This raises questions about the statistical significance of these results. At the moment, this issue is not discussed or even mentioned by the authors.  For transparency, I would suggest that the authors include the theoretical chance level in all tables and figures.
- The MI dataset used is relatively unknown (only cited once on Google Scholar), which does not make it a good benchmark. As this is the only MI dataset used, I believe it is necessary to conduct additional experiments on another, more common, MI benchmark.
- Line 124, the authors point out that few discussions were made on how to fine-tune models to downstream tasks in the BCI literature. While it is true that there are few, they are not nonexistent. As this is the main topic of the article, the few works that were done in that direction should at least be reported, if not compared to. The following two studies compared different downstream architectures combined with different fine-tuning regimes. In particular, they both explored additive fine-tuning algorithms, which is in contradiction with the statement line 145.
    - Kostas et al. (2021) https://doi.org/10.3389/fnhum.2021.653659
    - Guetschel et al. (2024) https://doi.org/10.3217/978-3-99161-014-4-003
- The method proposed by the authors can only be applied to transformer-based pre-trained models and requires doing “surgical” modifications to the architecture. This is not easy to implement compared to simple finetuning.
- The appendix is missing.

**Questions:**

- Why not evaluate on a more commonly used benchmark such as dataset B from 2008 BCI competition? https://www.doi.org/10.1109/TNSRE.2007.906956
- Line 322: The term “fune-tuned” is confusing in this context, it suggests that the supervised methods are pre-trained. Is it the case?
- Could you include the training times of the different methods?

---

### Note · Authors · 2024-11-19

I have read and agree with the venue's withdrawal policy on behalf of myself and my co-authors.